# GRAPH PARSING NETWORKS

**Yunchong Song**[1], **Siyuan Huang**[1], **Xinbing Wang**[1], **Chenghu Zhou**[2], **Zhouhan Lin**[1]*
[1]Shanghai Jiao Tong University,   [2]Chinese Academy of Sciences
{ycsong, xwang8}@sjtu.edu.cn, lin.zhouhan@gmail.com

## ABSTRACT

Graph pooling compresses graph information into a compact representation. State-of-the-art graph pooling methods follow a hierarchical approach, which reduces the graph size step-by-step. These methods must balance memory efficiency with preserving node information, depending on whether they use node dropping or node clustering. Additionally, fixed pooling ratios or numbers of pooling layers are predefined for all graphs, which prevents personalized pooling structures from being captured for each individual graph. In this work, inspired by bottom-up grammar induction, we propose an efficient graph parsing algorithm to infer the pooling structure, which then drives graph pooling. The resulting Graph Parsing Network (GPN) adaptively learns personalized pooling structure for each individual graph. GPN benefits from the discrete assignments generated by the graph parsing algorithm, allowing good memory efficiency while preserving node information intact. Experimental results on standard benchmarks demonstrate that GPN outperforms state-of-the-art graph pooling methods in graph classification tasks while being able to achieve competitive performance in node classification tasks. We also conduct a graph reconstruction task to show GPN's ability to preserve node information and measure both memory and time efficiency through relevant tests.[1]

## 1 INTRODUCTION

Graph neural networks (GNNs) (Scarselli et al., 2008; Bruna et al., 2013; Defferrard et al., 2016; Kipf & Welling, 2017; Veličković et al., 2018; Hamilton et al., 2017; Gilmer et al., 2017; Xu et al., 2019) have shown remarkable success in processing graph data from various fields (Li & Goldwasser, 2019; Yan et al., 2019; Suárez-Varela et al., 2021; Kipf et al., 2018). Graph pooling is built on top of GNNs. It aims to capture graph-level information by compressing a set of nodes and their underlying structure into a more concise representation. Early graph pooling methods, such as mean or add pool (Atwood & Towsley, 2016; Xu et al., 2019), perform permutation-invariant operations on all nodes in a graph. These flat pooling methods disregard the distinctions between nodes and fail to model the latent hierarchical structure within a graph. Thus, they can only capture information at the graph level in a rough manner. To overcome this limitation, researchers drew inspiration from pooling designs in computer vision (Ronneberger et al., 2015; Noh et al., 2015; Jégou et al., 2017), they turned to perform hierarchical pooling on graph data to capture structural information more accurately. The dominant approaches are node dropping (Gao & Ji, 2019; Lee et al., 2019) and node clustering (Ying et al., 2018; Bianchi et al., 2020). Each approach has its own advantages and disadvantages: node dropping methods reduce the size of a graph by dropping a certain proportion of nodes at each step. While they are memory-efficient, they irreversibly lose node information and may damage graph connectivity (Baek et al., 2021; Wu et al., 2022), leading to sub-optimal performance. On the other hand, node clustering methods perform clustering at each pooling layer, allowing for soft cluster assignments for each node, thus preserving its information. However, this comes at the cost of a dense cluster assignment matrix with quadratic memory complexity (Baek et al., 2021; Liu et al., 2023), sacrificing scalability. The comparison of graph pooling methods can refer to Figure 1. Hierarchical pooling methods need to strike a balance between preserving node information and being memory efficient, while our model excels in both aspects.

---

*Zhouhan Lin is the corresponding author.
[1]Code is available at https://github.com/LUMIA-Group/GraphParsingNetworks

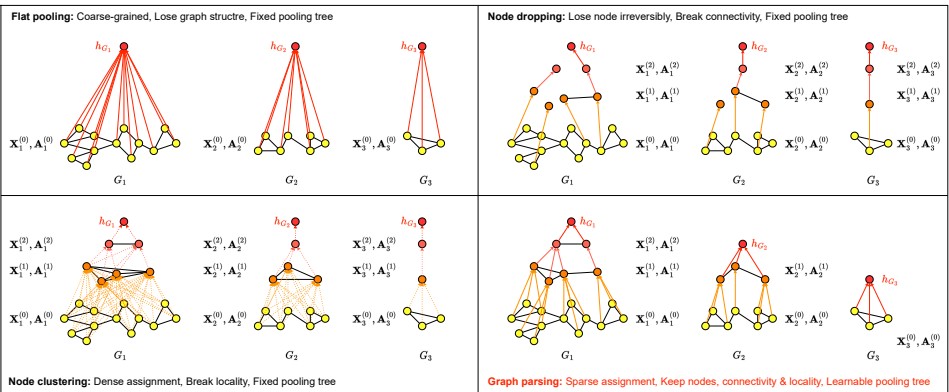

Figure 1: Comparison of graph pooling methods. The colors of the nodes indicate their pooling levels. The solid/dotted lines with arrows represent hard/soft node assignment matrixes.

Another fundamental problem of existing graph pooling methods is that they are unable to learn personalized pooling structure for each individual graph (Liu et al., 2023), i.e., they predefine a fixed pooling ratio or number of layers for all graphs. However, some graphs may be more compressible compared to others and imposing a constant pooling ratio may hurt performance on less compressible ones (Gao & Ji, 2019; Ranjan et al., 2020; Wu et al., 2022), leading to suboptimal performance. We represent the graph pooling structure as a rooted-tree to illustrate the problem, which we name the *pooling tree* (cf. Figure 1). The root node represents the graph-level representation, while the leaf nodes are taken from the input graph. The nodes produced by the $k$-th pooling layer correspond to the height $k$ of the tree. Ideally, the model should learn a personalized pooling tree for each graph with a flexible shape. This means that we should not constrain the total height $h_{max}$ or the number of nodes $n_k$ at height $k$. There are numerous methods that focus on optimizing graph structure (Gasteiger et al., 2019; Jiang et al., 2019; Chen et al., 2020b; Jin et al., 2020; Suresh et al., 2021; Topping et al., 2022; Bo et al., 2023). However, only a few (Diehl, 2019; Wu et al., 2022) attempt to optimize pooling structure, and even those can only learn a partially flexible shape of the pooling tree and not in an end-to-end fashion. In Appendix B and Appendix C, we discuss these methods in detail and highlight the differences. Our model is the first to end-to-end learning of a fully flexible pooling tree.

To overcome these two crucial problems, we propose a graph parsing algorithm inspired by recent advancements in bottom-up grammar induction (Shen et al., 2018b;a; 2019). In natural language processing, grammar induction techniques extract latent discrete structures from input text sequences without any guided supervision. We adapted this approach for the graph domain and used it to learn pooling trees. As shown in Figure 2, our model employs a graph parsing algorithm that assigns a discrete assignment to each node at layer $k$, allowing us to *infer* the number of nodes $n_k$ at height $k$. The parsing process ends when the graph has been fully parsed (i.e., the graph size remains unchanged), enabling us to *infer* the total height $h_{max}$. Therefore, our model can learn a pooling tree with a flexible shape for each individual graph in an end-to-end fashion. Our parsing algorithm is based on locality, thus ensures graph connectivity during pooling. We clustering nodes instead of dropping them, which preserves node information intact, and the discrete clustering process guarantees memory efficiency. Jo et al. (2021) also performs edge-centric graph pooling, however, they still need to pre-define the number of pooling layers or pooling ratio. A more detailed introduction for related works can be found in Appendix B.

## 2 PROPOSED METHOD

In this section, we first divide the graph pooling layer into three modules and introduce our design for each. Then, we present the details of our proposed graph parsing algorithm. Finally, we introduce the model architecture that is applied to both graph classification and node classification tasks.

### 2.1 GRAPH POOLING COMPONENTS

We represent a graph $G = (\mathcal{V}, \mathcal{E}), |\mathcal{V}| = n, |\mathcal{E}| = m$ with its feature matrix $\mathbf{X} \in \mathbb{R}^{n \times d}$ and adjacent matrix $\mathbf{A} \in \mathbb{R}^{n \times n}$, where $n, m, d$ represents the number of nodes, the number of edges and the

dimension of the node feature vector, respectively. Hierarchical graph pooling compress the original graph $G^{(0)} = G$ into a single vector, step-by-step. At each pooling layer, an input graph $G^{(k)}$, which has $n^{(k)}$ nodes, is transformed into a smaller output graph $G^{(k+1)}$ with fewer nodes ($n^{(k+1)} < n^{(k)}$). In this study, we present a unified framework which consists of three modules: graph information encoding, structural transformation, and multiset computation (see Figure 2).

**Graph information encoding** This module extracts node features and graph structure to generate useful metrics. For instance, it can generate node scores for node dropping methods or cluster assignment vectors for node clustering methods. Considerable effort has been put into improving the quality of these metrics, as evidenced by the advanced designs discussed in Section B. To encode structural information, we apply a GNN block and then use an MLP on adjacent nodes to calculate the edge score:

$$\mathbf{H}^{(k)} = \text{GNN}\left(\mathbf{X}^{(k)}, \mathbf{A}^{(k)}\right), \mathbf{C}_{i,j}^{(k)} = \sigma\left(\text{MLP}\left(\mathbf{H}_{i,:}^{(k)} \odot \mathbf{H}_{j,:}^{(k)}\right)\right) \tag{1}$$

$\sigma(\cdot)$ here is a sigmoid function, $\mathbf{C}^{(k)}$ is the edge score matrix that satisfy $\mathbf{C}^{(k)} = \mathbf{C}^{(k)} \odot \mathbf{A}^{(k)}$. In practice, we simply implement the GNN block with multiple GCN layers.

**Structural transformation** This module consists of two steps: node assignment and output graph construction. In the $k$-th pooling layer, the first step involves mapping nodes from the input graph to the output graph, resulting in an assignment matrix $\mathbf{S}^{(k)} \in \mathbb{R}^{n^{(k)} \times n^{(k+1)}}$. $\mathbf{S}^{(0)}, \mathbf{S}^{(1)}, \cdots, \mathbf{S}^{(K)}$ fully encode the pooling tree with height $K$. The primary challenge here is to learn a flexible pooling tree for each graph, as we described in Section 1. We propose a graph parsing algorithm $\mathcal{A}$ to construct node assignment and perform matrix multiplication for the second step:

$$\mathbf{S}^{(k)} = \mathcal{A}\left(\mathbf{C}^{(k)}\right), \mathbf{A}^{(k+1)} = \mathbf{S}^{(k)^{\text{T}}} \mathbf{A}^{(k)} \mathbf{S}^{(k)} \tag{2}$$

**Multiset computation** This module computes the feature matrix $\mathbf{X}^{(k+1)}$ for $G^{(k+1)}$. As the input node set can be treated as a multiset, this module can be seen as a multiset function (Baek et al., 2021). Simple implementations include operators such as $\text{ADD}(\cdot), \text{MEAN}(\cdot)$, and $\text{MAX}(\cdot)$. Recent studies have explored more expressive methods like DeepSets (Zaheer et al., 2017), or attention mechanisms (Baek et al., 2021). We implement this module using DeepSets:

$$\hat{\mathbf{X}}^{(k+1)} = \text{DeepSets}\left(\mathbf{H}^{(k)}, \mathbf{S}^{(k)}\right) = \text{MLP}_2\left(\mathbf{S}^{(k)^{\text{T}}} \text{MLP}_1\left(\mathbf{H}^{(k)}\right)\right) \tag{3}$$

To update the MLP in Equation 1, we need to involve edge scores in backpropagation since the graph parsing algorithm $\mathcal{A}$ is not differentiable. To achieve this, we compute a mask $\mathbf{y}^{(k)}$ through edge scores. Let us consider a subgraph $\hat{G}_i^{(k)} \subseteq G^{(k)}$ induced by cluster $v_i$ in $G^{(k+1)}$. We denote edges inside $\hat{G}_i^{(k)}$ as $\hat{\mathcal{E}}_i^{(k)}$, and $\mathbf{1}$ is a $d$-dimension ones vector. Through adding the score and then multiplying to the features, the cluster can be aware of the number of edges assigned to it.[2]

$$\mathbf{X}^{(k+1)} = \hat{\mathbf{X}}^{(k+1)} \odot \left(\mathbf{y}^{(k)} \mathbf{1}^{\text{T}}\right), \mathbf{y}_i^{(k)} = \text{ADD}\left(\left\{\mathbf{C}_{j,k}^{(k)} | (j,k) \in \hat{\mathcal{E}}_i^{(k)}\right\}\right) \tag{4}$$

## 2.2 GRAPH PARSING ALGORITHM

Our graph parsing algorithm $\mathcal{A}$ takes edge score matrix $\mathbf{C}^{(k)}$ as input and returns an assignment matrix $\mathbf{S}^{(k)}$. We do not impose any constraints on the pooling ratio or the number of clusters. Therefore, we infer clusters from nodes in a manner similar to how larger units (e.g., clauses) are inferred from smaller units (e.g., phrases) in grammar induction. In order to make the pooling structure learnable, we rely on both the graph topology and a set of continuous values defined on it, namely edge scores. As such, when edge scores are updated through gradients, the pooling structure can also be updated via decoding from it. To elaborate on this decoding process, we first introduce three operators that are utilized within it.

**Definition 1.** $\text{DOM}(\cdot)$ *(short for "dominant"): this operator selects the dominant edge for each node, which determines its assignment. First it performs a row-wise max indexing on the edge score matrix $\mathbf{C}^{(k)}$ to obtain seed edges $idx$, then construct matrix $\hat{\mathbf{C}}^{(k)}$ by filtering out other edges:*

$$idx = \text{argmax}_{\text{row-wise}}(\mathbf{C}^{(k)}), \hat{\mathbf{C}}_{i,j}^{(k)} = \begin{cases} \mathbf{C}_{i,j}^{(k)}, (i,j) \in idx \\ 0, otherwise \end{cases} \tag{5}$$

---

[2]Details for Equation 4 and end-to-end training can be found in Appendix F.

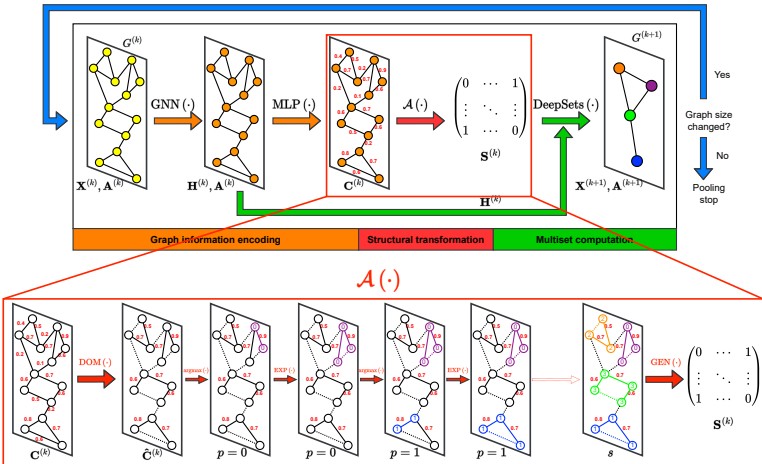

Figure 2: Proposed graph parsing networks, driven by the graph parsing algorithm $\mathcal{A}$. The box with three stages indicates the graph pooling layer, which recurrently builds the entire model. The red box shows the details and an example of how $\mathcal{A}$ works.

**Definition 2.** $\mathrm{EXP}(\cdot)$ *(short for "expand"): this operator expands a set of seed edges $idx$ on the filtered graph $\hat{\mathbf{C}}^{(k)}$ through a neighbor lookup. First induce the node set $l$ using $idx$, then merge neighboring edges $idx'$ that are dominated by them. $idx_x, idx_y$ represent the sets of starting and ending node indices for the edge set $idx$:*

$$l = \mathrm{union}\left(\{idx_x\}, \{idx_y\}\right), idx' = \{(i,j)|(i \in l \vee j \in l) \wedge \hat{\mathbf{C}}^{(k)}_{i,j} \neq 0\} \tag{6}$$

**Definition 3.** $\mathrm{GEN}(\cdot)$ *(short for "generate"): given an assignment mapping $s$ that maps node $i$ to cluster $j$, this operator generates the assignment matrix $\mathbf{S}^{(k)}$ using $s$:*

$$\mathbf{S}^{(k)}_{i,j} = \begin{cases} 1, (i,j) \in s \\ 0, otherwise \end{cases} \tag{7}$$

Based on these definitions, we present the aforementioned decoding process in Algorithm 1. To facilitate understanding, we divided it into three stages. In Stage 1, we use the $\mathrm{DOM}(\cdot)$ operator to identify the dominant edge for each node, which determines its assignment. To ensure *memory efficient*, we opt for a sparse assignment by selecting a single dominant edge for each node. This also protects *locality* by avoiding instances where a single node is assigned to multiple clusters, which could result in unconnected nodes being assigned to the same cluster. During this stage, we initialize an empty mapping $s$ that associates nodes with clusters. As we create clusters and map nodes to them one by one, we also set up an indicator variable $p$, which keeps track of the latest cluster's index.

In Stage 2, we use $\hat{\mathbf{C}}^{(k)}$ to create $s$. This is done through a double-loop approach, where new cluster is generated in the outer loop and nodes are assigned to it in the inner loop. In the outer loop, we choose the edge with the highest

---

**Algorithm 1** Graph parsing algorithm $\mathcal{A}$

**Input:** edge score matrix $\mathbf{C}^{(k)}$
**Output:** assignment matrix $\mathbf{S}^{(k)}$

1: $\hat{\mathbf{C}}^{(k)} \leftarrow \mathrm{DOM}\left(\mathbf{C}^{(k)}\right)$     ▷ **Stage 1**
2: $s \leftarrow \emptyset$
3: $p \leftarrow 0$
4: **while** $\mathrm{sum}(\hat{\mathbf{C}}^{(k)}) \neq 0$ **do**     ▷ **Stage 2**
5:     $idx \leftarrow \mathrm{argmax}(\hat{\mathbf{C}}^{(k)})$
6:     **do**
7:         $q \leftarrow |idx|$
8:         $l, idx' \leftarrow \mathrm{EXP}\left(idx, \hat{\mathbf{C}}^{(k)}\right)$
9:         $idx \leftarrow \mathrm{union}\left(idx, idx'\right)$
10:    **while** $|idx| = q$
11:    $s \leftarrow \mathrm{union}\left(s, \{(i,p)|i \in l\}\right)$
12:    $\hat{\mathbf{C}}^{(k)}_{i,j} \leftarrow 0, \forall (i,j) \in idx$
13:    $p \leftarrow p + 1$
14: **end while**
15: $\mathbf{S}^{(k)} = \mathrm{GEN}(s)$     ▷ **Stage 3**
16: **return** $\mathbf{S}^{(k)}$

---

score in $\hat{\mathbf{C}}^{(k)}$ as the seed edge. This edge is considered as the first member of a new cluster. In the inner loop, we expand this seed edge iteratively using the $\mathrm{EXP}(\cdot)$ operator which performs neighborhood lookup and merges. The inner loop terminates when no additional edges can be covered by expanding from the seed edges, i.e., $|idx| = q$. We then map all the nodes in final node set $l$ to the same $p$-th cluster. After that, we clear all the edges covered in this loop ($idx$) from $\hat{\mathbf{C}}^{(k)}$,

and update the indicator $p$. In Stage 3, we use the $\text{GEN}(\cdot)$ operator to create an assignment matrix $\mathbf{S}^{(k)}$ with $s$. If the graph has isolated nodes, we generate a separate cluster for each isolated node and create a corresponding mapping in $s$.

**Complexity** The time complexity for Algorithm 1 is $\mathcal{O}(\sum_{i=1}^{n^{(k+1)}} d_i)$, $d_i$ indicate the diameter of the subgraph $\hat{G}_i^{(k)} \subseteq G^{(k)}$ induced by node $v_i$ in $G^{(k+1)}$. Compared to EdgePool, which has a time complexity of $\mathcal{O}(m^{(k)})$, our worst case time complexity of $\mathcal{O}(n^{(k)})$ is much smaller. In Appendix A.3, we provide further comparison to SEP and demonstrate that our model is more efficient than it. These results indicate that our model is also *time efficient* and can handle dense or large graphs.

**Proposition 4.** *(Proof in Appendix A.1.) In $k$-th graph pooling layer, the time complexity for Algorithm 1 is $\mathcal{O}(\sum_{i=1}^{n^{(k+1)}} d_i)$, which is upper-bounded by the worst case $\mathcal{O}(n^{(k)})$.*

**Permutation invariance** Permutation invariance is a crucial property in graph representation learning as it ensures the model's ability to generalize on graph data. As the other components (GNN, DeepSets) of our model are already permutation invariant, we only focus on analyzing the permutation invariance of the graph parsing algorithm $\mathcal{A}$.

**Proposition 5.** *(Proof in Appendix A.2.) Given a permutation matrix $\mathcal{P}$, suppose the row-wise non-zero entries in $\mathbf{C}$ are distinct, $\mathcal{A}(\mathbf{C}) = \mathcal{A}(\mathcal{P}\mathbf{C}\mathcal{P}^{\mathrm{T}})$, thus $\mathcal{A}$ is permutation invariant.*

**Graph connectivity** Keep the graph connectivity during pooling is important. Our parsing algorithm keep the connectivity of the input graph unchanged during graph pooling, which guarantees a fluent flow of information on the pooled graph.

**Proposition 6.** *(Proof in Appendix A.4.) For nodes $i, j \in G^{(k)}$, if there is a path $s_{ij}^{(k)}$ connect them, then a path $s_{p_i p_j}^{(k+1)}$ would exist between the corresponding clusters $p_i, p_j \in G^{(k+1)}$ after parsing.*

## 2.3 Model Architecture

**Graph-level architecture** By utilizing the pooling layer that was introduced in previous sections, we can directly employ this model for graph classification tasks. The architecture of our model (depicted in Figure 3) comprises solely of one pooling layer which progressively decreases the size of the input graph until it matches the output graph produced by our graph parsing algorithm $\mathcal{A}$. Subsequently, a multiset computation module is utilized to convert the final graph into a single node and generate a compressed vector for graph classification.

**Node-level architecture** We developed an encoder-decoder architecture (refer to Figure 3) for the node classification task, inspired by previous graph pooling methods used in node-level tasks like TopKPool (Gao & Ji, 2019) and SEP (Wu et al., 2022). Our approach involves compressing the graph into a smaller size using bottom-up pooling in the encoder block, followed by expanding it in the decoder block. We store the assignment at every pooling layer during the encoding process and use this information for un-pooling in the decoding process. We implement $k$-th un-pooling layer as follows:

$$\mathbf{A}^{(k+1)} = \mathbf{S}^{\mathrm{T}^{(h-k)}} \mathbf{A}^{(k)} \mathbf{S}^{(h-k)} \quad (8)$$
$$\mathbf{X}^{(k+1)} = \mathbf{S}^{(h-k)} \mathbf{X}^{(k)} \quad (9)$$

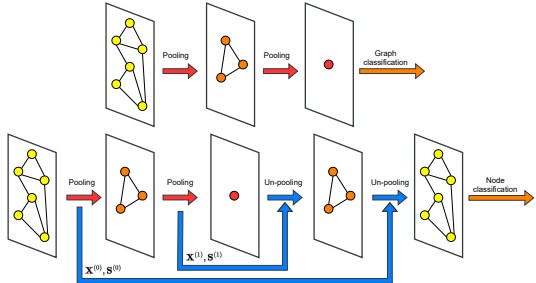

Figure 3: Graph-level and node-level GPN architecture. Each un-pooling layer receives an assignment matrix and node feature through a skip-connection from its corresponding pooling layer.

the variable $h$ represents the height of the pooling tree in the encoder block. The assignment matrix $\mathbf{S}^{(h-k)}$ is stored in the $(h-k)$-th pooling layer and corresponds to the $k$-th un-pooling layer. Additionally, skip connections are applied between the encoder and decoder, following methods used in Gao & Ji (2019); Wu et al. (2022).

Table 1: Graph classification accuracy with mean and standard deviation based on 20 random seeds, we bold the model with best performance.

| | DD | PROTEINS | NCI1 | NCI109 | FRANKENSTEIN |
|---|---|---|---|---|---|
| # Graphs | 1,178 | 1,113 | 4,110 | 4,127 | 4,337 |
| # Nodes (Avg.) | 284.3 | 39.1 | 29.9 | 29.7 | 16.9 |
| # Classes | 2 | 2 | 2 | 2 | 2 |
| Set2Set (Vinyals et al., 2016) | $71.60_{\pm 0.87}$ | $72.16_{\pm 0.43}$ | $66.97_{\pm 0.74}$ | $61.04_{\pm 2.69}$ | $61.46_{\pm 0.47}$ |
| Attention (Li et al., 2016) | $71.38_{\pm 0.78}$ | $71.87_{\pm 0.60}$ | $69.00_{\pm 0.49}$ | $67.87_{\pm 0.40}$ | $61.31_{\pm 0.41}$ |
| GCN (Kipf & Welling, 2017) | $68.33_{\pm 1.30}$ | $73.83_{\pm 0.33}$ | $73.92_{\pm 0.43}$ | $72.77_{\pm 0.57}$ | $60.47_{\pm 0.62}$ |
| SortPool (Zhang et al., 2018) | $71.87_{\pm 0.96}$ | $73.91_{\pm 0.72}$ | $68.74_{\pm 1.07}$ | $68.59_{\pm 0.67}$ | $63.44_{\pm 0.65}$ |
| GMT (Baek et al., 2021) | $\mathbf{78.11_{\pm 0.76}}$ | $74.97_{\pm 0.79}$ | $70.35_{\pm 0.65}$ | $69.45_{\pm 0.45}$ | $66.76_{\pm 0.60}$ |
| TopKPool (Gao & Ji, 2019) | $75.01_{\pm 0.86}$ | $71.10_{\pm 0.90}$ | $67.02_{\pm 2.25}$ | $66.12_{\pm 1.60}$ | $61.46_{\pm 0.84}$ |
| SAGPool (Lee et al., 2019) | $76.45_{\pm 0.97}$ | $71.86_{\pm 0.97}$ | $67.45_{\pm 1.11}$ | $67.86_{\pm 1.41}$ | $61.73_{\pm 0.76}$ |
| HGP-SL (Zhang et al., 2019) | $75.16_{\pm 0.69}$ | $74.09_{\pm 0.84}$ | $75.97_{\pm 0.40}$ | $74.27_{\pm 0.60}$ | $63.80_{\pm 0.50}$ |
| GSAPool (Zhang et al., 2020) | $76.07_{\pm 0.81}$ | $73.53_{\pm 0.74}$ | $70.98_{\pm 1.20}$ | $70.68_{\pm 1.04}$ | $60.21_{\pm 0.69}$ |
| ASAP (Ranjan et al., 2020) | $76.87_{\pm 0.70}$ | $74.19_{\pm 0.79}$ | $71.48_{\pm 0.42}$ | $70.07_{\pm 0.55}$ | $66.26_{\pm 0.47}$ |
| SPGP (Lee et al., 2022) | $77.76_{\pm 0.73}$ | $75.20_{\pm 0.74}$ | $78.90_{\pm 0.27}$ | $77.27_{\pm 0.32}$ | $68.92_{\pm 0.71}$ |
| DiffPool (Ying et al., 2018) | $66.95_{\pm 2.41}$ | $68.20_{\pm 2.02}$ | $62.32_{\pm 1.90}$ | $61.98_{\pm 1.98}$ | $60.60_{\pm 1.62}$ |
| EdgePool (Diehl, 2019) | $72.77_{\pm 3.99}$ | $71.73_{\pm 4.31}$ | $78.93_{\pm 2.22}$ | $76.79_{\pm 2.39}$ | $69.42_{\pm 2.28}$ |
| MinCutPool (Bianchi et al., 2020) | $77.36_{\pm 0.49}$ | $75.00_{\pm 0.96}$ | $75.88_{\pm 0.57}$ | $74.56_{\pm 0.47}$ | $68.39_{\pm 0.40}$ |
| HoscPool (Duval & Malliaros, 2022) | $76.83_{\pm 0.90}$ | $75.05_{\pm 0.81}$ | $78.69_{\pm 0.45}$ | $78.36_{\pm 0.52}$ | $68.37_{\pm 0.61}$ |
| SEP (Wu et al., 2022) | $76.54_{\pm 0.99}$ | $75.28_{\pm 0.65}$ | $78.09_{\pm 0.47}$ | $76.48_{\pm 0.48}$ | $67.79_{\pm 1.54}$ |
| **GPN** | $77.09_{\pm 0.81}$ | $\mathbf{75.62_{\pm 0.74}}$ | $\mathbf{80.87_{\pm 0.43}}$ | $\mathbf{79.20_{\pm 0.47}}$ | $\mathbf{70.45_{\pm 0.53}}$ |

An informative node representation is crucial for node-level tasks. To achieve this, we utilize a separate GNN encoder for the multiset computation module. In the node-level architecture, Equation 3 in the pooling layer needs to be rewritten as:

$$\hat{\mathbf{X}}^{(k+1)} = \text{DeepSets}\left(\text{GNN}'\left(\mathbf{X}^{(k)}, \mathbf{A}^{(k)}\right), \mathbf{S}^{(k)}\right) \tag{10}$$

## 3 EXPERIMENTS

In this section, we evaluate our model's performance on six graph classification and four node classification datasets. We also demonstrate how the node information is well-preserved through a graph reconstruction task. To assess the efficiency of our parsing algorithm, we conduct time and memory tests. Additionally, we showcase our model's ability to learn a personalized pooling structure for each graph by visualizing the number of pooling layers during training across different datasets. Finally, an ablation study is performed on all three graph pooling modules to determine their contributions.

### 3.1 GRAPH CLASSIFICATION

Graph classification requires a model to classify a graph into a label, which calls for graph-level representation learning.

**Experimental setup** We evaluate our model on five widely-used graph classification benchmarks from TUDatasets (Morris et al., 2020): DD, PROTEINS, NCI1, NCI109, and FRANKENSTEIN. Table 1 presents the statistics of them. Additionally, we evaluate our model's performance at scale on ogbg-molpcba, one of the largest graph classification datasets in OGB (Hu et al., 2020), comprising over 400,000 graphs. For baselines, we select GCN (Kipf & Welling, 2017) (with simple mean pooling), Set2Set (Vinyals et al., 2016), Global-Attention (Li et al., 2016), SortPool (Zhang et al., 2018), and GMT (Baek et al., 2021) from the flat-pooling family. For hierarchical pooling methods, we include TopKPool (Gao & Ji, 2019), SAGPool (Lee et al., 2019), HGP-SL (Zhang et al., 2019), GSAPool (Zhang et al., 2020), ASAP (Ranjan et al., 2020), and SPGP (Lee et al., 2022) for node dropping and DiffPool (Ying et al., 2018), EdgePool (Diehl, 2019), MinCutPool (Bianchi et al., 2020), HoscPool (Duval & Malliaros, 2022), and SEP (Wu et al., 2022) for node clustering[3]. We adopt 10-fold cross-validation with 20 random seeds for both model initialization and fold generation. This

---

[3]SEP is a special case in node clustering based pooling, since it computes a sparse pooling tree before training, while others generate dense assignments for each forward pass.

Table 3: Node classification accuracy with mean and standard deviation based on 10 splits, we bold the model with best performance.

|  | Cora | CiteSeer | PubMed | Film |
|---|---|---|---|---|
| # Nodes | 2,708 | 3,327 | 18,717 | 7,600 |
| # Edges | 5,278 | 4,676 | 44,327 | 26,752 |
| # Classes | 6 | 7 | 3 | 5 |
| MLP | $75.69_{\pm 2.00}$ | $74.02_{\pm 1.90}$ | $87.16_{\pm 0.37}$ | $36.53_{\pm 0.70}$ |
| GCN (Kipf & Welling, 2017) | $86.98_{\pm 1.27}$ | $76.50_{\pm 1.36}$ | $88.42_{\pm 0.50}$ | $27.32_{\pm 1.10}$ |
| GraphSAGE (Hamilton et al., 2017) | $86.90_{\pm 1.04}$ | $76.04_{\pm 1.30}$ | $88.45_{\pm 0.50}$ | $34.23_{\pm 0.99}$ |
| GAT (Veličković et al., 2018) | $86.33_{\pm 0.48}$ | $76.55_{\pm 1.23}$ | $87.30_{\pm 1.10}$ | $27.44_{\pm 0.89}$ |
| MixHop (Abu-El-Haija et al., 2019) | $87.61_{\pm 0.85}$ | $76.26_{\pm 1.33}$ | $85.31_{\pm 0.61}$ | $32.22_{\pm 2.34}$ |
| PairNorm (Zhao & Akoglu, 2020) | $85.79_{\pm 1.01}$ | $73.59_{\pm 1.47}$ | $87.53_{\pm 0.44}$ | $27.40_{\pm 1.24}$ |
| Geom-GCN (Pei et al., 2020) | $85.35_{\pm 1.57}$ | $\mathbf{78.02_{\pm 1.15}}$ | $89.95_{\pm 0.47}$ | $31.59_{\pm 1.15}$ |
| GCNII (Chen et al., 2020a) | $\mathbf{88.37_{\pm 1.25}}$ | $77.33_{\pm 1.48}$ | $\mathbf{90.15_{\pm 0.43}}$ | $37.44_{\pm 1.30}$ |
| H2GCN (Zhu et al., 2020) | $87.87_{\pm 1.20}$ | $77.11_{\pm 1.57}$ | $89.49_{\pm 0.38}$ | $35.70_{\pm 1.00}$ |
| GPRGNN (Chien et al., 2021) | $87.95_{\pm 1.18}$ | $77.13_{\pm 1.67}$ | $87.54_{\pm 0.38}$ | $34.63_{\pm 1.22}$ |
| GGCN (Yan et al., 2022) | $87.95_{\pm 1.05}$ | $77.14_{\pm 1.45}$ | $89.15_{\pm 0.37}$ | $37.54_{\pm 1.56}$ |
| NSD (Bodnar et al., 2022) | $87.30_{\pm 1.15}$ | $77.14_{\pm 1.85}$ | $89.49_{\pm 0.40}$ | $37.81_{\pm 1.15}$ |
| TopKPool (Gao & Ji, 2019) | $79.65_{\pm 2.82}$ | $71.97_{\pm 3.54}$ | $85.47_{\pm 0.71}$ | — |
| SEP (Wu et al., 2022) | $87.20_{\pm 1.12}$ | $75.34_{\pm 1.20}$ | $87.5_{\pm 0.35}$ | — |
| **GPN** | $88.07_{\pm 0.87}$ | $77.02_{\pm 1.65}$ | $89.61_{\pm 0.34}$ | $\mathbf{38.11_{\pm 1.23}}$ |

means there are 200 runs in total behind each data point, ensuring a fair comparison. This approach is consistent with Lee et al. (2019), Ranjan et al. (2020), and Lee et al. (2022). More details about datasets, implementations and model tuning can be found in Appendix H.1.

**Classification results** Table 1 shows that our model outperforms SOTA graph pooling methods on four out of five datasets, demonstrating its ability to capture graph-level information precisely. The sub-optimal performance of our model on the DD dataset can be attributed to graph size, as DD has an average node count of 284.3, making it the dataset with large and sparse graphs that pose long-range issues (Alon & Yahav, 2020). Resolving such issue requires global information, the complete attention graph in GMT (Baek et al., 2021) provide such information shortcut, thus they perform better. Notably, our method does not include these designs, yet it has achieved considerable results on DD. On the large-scale ogbg-molpcba dataset, our model outperforms all SOTA graph pooling models and significantly improves upon GCN's performance, demonstrating the superior scalability of our model, and the potential to handle complex large scale data.

Table 2: Graph classification accuracy on large scale dataset ogbg-molpcba.

|  | ogbg-molpcba |
|---|---|
| # Graphs | 437,929 |
| # Nodes (Avg.) | 26.0 |
| # Classes | 2 |
| GCN (Kipf & Welling, 2017) | $20.20_{\pm 0.24}$ |
| HGP-SL (Zhang et al., 2019) | $18.64_{\pm 0.28}$ |
| EdgePool (Diehl, 2019) | $23.48_{\pm 0.41}$ |
| ASAP (Ranjan et al., 2020) | OOM |
| GMT (Baek et al., 2021) | $23.70_{\pm 0.50}$ |
| SPGP (Lee et al., 2022) | $23.95_{\pm 0.97}$ |
| **GPN** | $\mathbf{26.65_{\pm 0.31}}$ |

## 3.2 NODE CLASSIFICATION

The node classification task focuses on node-level information. Each graph node is assigned a label, and the goal is to predict its label.

**Experimental setup** We conduct node classification experiments under full supervision for four datasets: Cora, CiteSeer, and PubMed from Sen et al. (2008), and Film from Pei et al. (2020). Table 3 shows the datasets' statistics. To evaluate our model's performance comprehensively, we compare it with a range of baselines, including classical methods: GCN (Kipf & Welling, 2017), GAT (Veličković et al., 2018), GraphSAGE (Hamilton et al., 2017), PairNorm (Zhao & Akoglu, 2020), MixHop (Abu-El-Haija et al., 2019); and SOTA methods: Geom-GCN (Pei et al., 2020), GCNII (Chen et al., 2020a), GPRGNN (Chien et al., 2021), H2GCN (Zhu et al., 2020), GGCN (Yan et al., 2022), and NSD (Bodnar et al., 2022). We further compare with two graph pooling methods, TopKPool (Gao & Ji, 2019) and SEP (Wu et al., 2022), as other pooling models cannot scale to thousands of nodes or lack a node-level architecture. We follow the same dataset splitting and settings as Zhu et al. (2020), Yan et al. (2022) and Bodnar et al. (2022) and report the average results of 10 dataset splits. Dataset and implementation details can be found in Appendix H.2.

**Classification results** The results in Table 3 demonstrate the competitiveness of our model on the selected datasets. Our approach achieves SOTA performance on Film dataset and performs comparably to the SOTA model for node classification, GCNII, on Cora and PubMed datasets. Furthermore, we surpass most baseline models on CiteSeer dataset. Since the compared SOTA methods are explicitly designed for node-level tasks and did not extend to graph-level, our overall performance is satisfactory. Notably, when compared with the leading graph pooling model SEP, we outperform it on all datasets. This reveals the superior generality across graph-level and node-level tasks of our proposed method.

### 3.3 GRAPH RECONSTRUCTION

To demonstrate our model's ability to maintain node information, we conduct a graph reconstruction experiment. This can visually illustrate the potential structural damage from graph pooling.

**Experimental setup** We follow Bianchi et al. (2020) to construct a graph autoencoder that includes pooling and un-pooling operations on the input graph $G = (\mathbf{X}, \mathbf{A})$ to recover the original node features. The recovered graph $G^r = (\mathbf{X}^r, \mathbf{A}^r)$ has the same structure as the input graph, but the node features are produced solely by the pooling and un-pooling process. The mean square error loss $|\mathbf{X} - \mathbf{X}^r|^2$ is used to train the autoencoder. We use the coordinates in a 2-D plane as the node features, thus the recovered node features can be visually assessed to evaluate the pooling model's ability to preserve node information. We test on the grid and ring graphs. We choose TopKPool and MinCutPool as baselines, which represent the node dropping and node clustering methods, respectively. More baseline results and higher resolution images can be found in Appendix H.3.

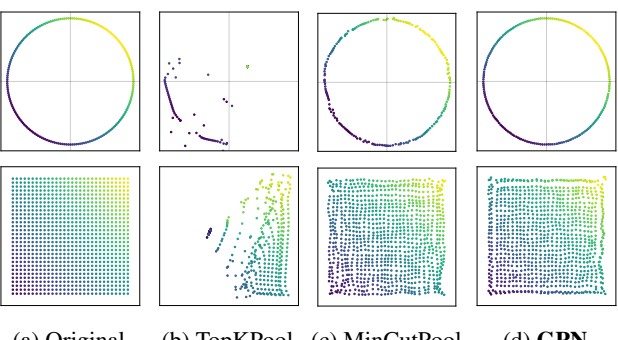

(a) Original    (b) TopKPool    (c) MinCutPool    (d) **GPN**

Figure 4: Graph reconstruction task on ring and grid synthetic graphs, which tests whether a graph pooling model can preserve node information intact.

**Reconstruction results** Figure 4 shows the graph reconstruction results on a 2D plane. It is obvious that TopKPool, which drops nodes, damages node information and performs poorly on two graphs. In contrast, MinCutPool uses clustering and preserves the original coordinates better. Compared to these methods, our model recovers a clearer ring graph and effectively preserves the central part of the grid graph, demonstrating its ability to preserve node information intact.

### 3.4 EFFICIENCY STUDY

In this section, we assess the memory and time efficiency of our proposed model. Since models from the same class will have similar memory and time performance, we choose one model from each class of pooling methods as our baseline. Specifically, we select GMT, TopKPool, MinCutPool, EdgePool for flat pooling, node dropping, node clustering, and edge-scoring based pooling respectively.

**Memory efficiency** We test memory efficiency as described in Baek et al. (2021). Varying numbers of nodes ($n$) were used to generate Erdos-Renyi graphs, while keeping the number of edges constant at $2n$ per graph. Applying pooling to each graph once, we compare our model's GPU memory usage with baselines (Figure 5). Our parsing-based model shows much better memory efficiency when compared to MinCutPool. This confirms our approach's memory-efficient nature.

**Time efficiency** To test time efficiency, we creat Erdos-Renyi graphs with $m = n^2/10$, and time one forward graph pooling as in Baek et al. (2021). Our results, as shown in Figure 6, demonstrate that GPN is much more efficient than EdgePool, a model also based on edge-scoring. This is because our model has an upper-bound time complexity of $\mathcal{O}(n)$, making it able to handle a large number of edges, whereas EdgePool's time complexity of $\mathcal{O}(m)$ would result in significant delays when processing too many edges.

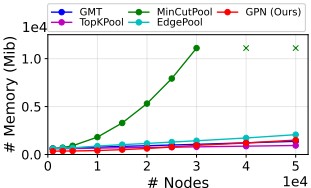

Figure 5: Memory efficiency test, "x" indicate the out-of-memory error.

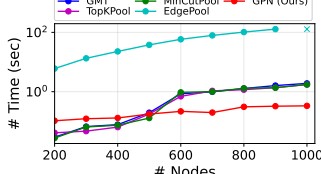

Figure 6: Time efficiency test, "x" indicate the out-of-memory error.

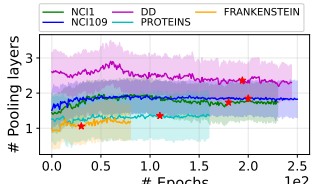

Figure 7: Number of pooling layers visualization during model training.

### 3.5 VISUALIZATION

In this section, we verify if our model learns personalized pooling structure for each graph. To achieve this, we have created a visualization in Figure 7 that shows the number of pooling layers learned during training on various datasets. Each data point and its shadow represent the mean and variance of the number of pooling layers for all graphs within a dataset. The star indicate the step with best validation loss. It is evident that the heights of star indicators vary across datasets, indicating our model learns distinct pooling layers for each. The fluctuating curve and its shadows during training confirm that our model effectively learns personalized pooling layers for every graph in a dataset.

### 3.6 ABLATION STUDY

In this subsection, we conducted an ablation study to assess the contribution of each module that we described in Section 2.1. We use four datasets, including both graph and node levels. We evaluated these variants: for the graph information encoding module, we used GAT or GIN layers instead of GCN layers; we transformed the input graph to a complete graph to check the necessity

Table 4: Ablation study on three components of a graph pooling layer, covering both graph dataset and node dataset.

|  | Graph-level | | Node-level | |
|---|---|---|---|---|
|  | **PROTEINS** | **FRANKENSTEIN** | **Cora** | **CiteSeer** |
| Original | $75.62_{\pm 0.74}$ | $70.45_{\pm 0.53}$ | $88.07_{\pm 0.87}$ | $77.02_{\pm 1.65}$ |
| w/GAT | $75.41_{\pm 0.91}$ | $69.94_{\pm 2.56}$ | $87.46_{\pm 1.13}$ | $77.16_{\pm 1.50}$ |
| w/GIN | $75.02_{\pm 1.25}$ | $70.18_{\pm 0.44}$ | $86.62_{\pm 1.26}$ | $76.50_{\pm 1.67}$ |
| w/Mean pool | $75.26_{\pm 1.13}$ | $69.75_{\pm 0.54}$ | $87.40_{\pm 1.02}$ | $76.50_{\pm 1.51}$ |
| w/Max pool | $75.12_{\pm 0.68}$ | $69.61_{\pm 0.46}$ | $87.08_{\pm 1.26}$ | $76.23_{\pm 1.29}$ |
| Complete graph | $74.12_{\pm 1.63}$ | $67.03_{\pm 1.05}$ | $74.85_{\pm 1.47}$ | OOM |

of graph locality and our proposed graph parsing algorithm; we tested two simple aggregators without parameters, mean pool and max pool, instead of DeepSets for multiset computation. Table 4 shows that overall superior performance was obtained by these variants, except when we broke the locality with a complete graph, the model suffered a significant performance drop, highlighting the importance of graph locality and underscoring the crucial role of our parsing algorithm. The good performance on other variants indicates that our model is robust regarding the selection of GNN encoder and multiset function. Notably, GPN with a GAT backbone get better results on CiteSeer, which reveals the potential of our model.

## 4 CONCLUSION

In this paper, we address two pressing challenges associated with hierarchical graph pooling: the inability to preserve node information while simultaneously maintaining a high degree of memory efficiency, and the lack of a mechanism for learning graph pooling structures for each individual graph. To address these challenges, we introduce a novel graph parsing algorithm. We demonstrate that our model outperforms existing state-of-the-art graph pooling methods in graph classification, while also delivering competitive performance in node classification. Additionally, a graph reconstruction task validates our model's ability to preserve node information intact, while an efficiency study confirms the effectiveness of our solution in terms of both time and memory consumption. All the proofs, a more detailed introduction to related works, the limitations of our model and broader impacts can be found in Appendix due to the space limit.

## 5 ETHICS STATEMENT

In this work, we propose a novel graph pooling model. Our model is a general graph pooling model that does not make explicit assumptions about data distribution, thus avoiding the occurrence of (social) bias. However, our data-driven pooling mechanism needs to learn from samples with labels, and it is possible to be misled by targeting corrupted graph structures or features, which can have a negative social impact because both internet data and social networks can be represented as graphs. We can develop corresponding ethical guidelines and constraints for the usage of our model to avoid such problems.

## 6 REPRODUCIBILITY STATEMENT

We provide an introduction to our used datasets and model configuration in the experimental section of our paper. The dataset is mentioned first in this section. In our Appendix, we report the hardware and software environment used in our experiments, the methods used for hyper-parameter tuning, and the steps for downloading, loading, and pre-processing the datasets. In our released source code, we list the steps required to reproduce the results and provide yaml files that include all the hyper-parameters for the reported performance in this paper.

## ACKNOWLEDGEMENT

This work was sponsored by the National Key Research and Development Program of China (No. 2023ZD0121402) and National Natural Science Foundation of China (NSFC) grant (No.62106143).

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

# A  PROOFS

## A.1  PROOFS REGRADING PROPOSITION 4

*Proof.* Since in the $k$-th graph pooling layer, the number of iterations for outer loop in Algorithm 1 is $n^{(k+1)}$, to prove the time complexity $\mathcal{O}(\sum_{i=1}^{n^{(k+1)}} d_i)$, we only need to prove the number of iterations in $i$-th inner loop is $d_i$. Notice that in Definition 2, the $\mathrm{DOM}(\cdot)$ performs a edges neighborhood lookup on graph $\hat{\mathbf{C}}^{(k)}$, and this lookup operation can be effectively parallelized for all edges, thus the number of iterations in inner loop equals to the lookup steps needed to traverse all the nodes in that subgraph (a subgraph corresponding to node $i$ in the output graph), which equals to the diameter $d_i$ of the subgraph.

Since graph $\hat{\mathbf{C}}^{(k)}$ has $n^{(k)}$ non-zero entries as we described in Definition 1, the time complexity can never exceed it; then, suppose the input graph $\mathbf{C}^{(k)}$ is a pure set of isolated nodes, without any structure, then the number of iterations in Algorithm 1 equals to $n^{(k)}$, which is the worst case. □

## A.2  PROOFS REGRADING PROPOSITION 5

*Proof.* Since the commonly used operations like $\mathrm{sum}(\cdot)$, $\mathrm{argmax}(\cdot)$, $\mathrm{union}(\cdot)$ in Algorithm 1 are based on set and are permutation invariant, we only need to prove 3 operators we introduced are also permutation invariant: for $\mathrm{DOM}(\cdot)$ operator, it first doing a row-wise max index, which is permutation invariant, then a masking is applied, since the masking is based on the idx from last step, thus it is also permutation invariant; for $\mathrm{EXP}(\cdot)$ operator, it performs a neighborhood lookup on matrix in a parallel way (has no need to specify the order), thus it is also permutation invariant; for $\mathrm{GEN}(\cdot)$ operator, it's results has no relationship with ordering since it's just a matrix filling process, thus it is also permutation invariant.

The above analysis are based on a set of distinct values in edge score matrix $\mathbf{C}^{(k)}$, if there are some duplicate values, the $\mathrm{argmax}(\cdot)$ operator cannot guarantee uniqueness. Thus we need to assume the row-wise non-zero entries in $\mathbf{C}$ are distinct, when the assumption breaks, it means there are at least two nodes with common neighbors and their node features are exactly the same. □

## A.3  PROOFS REGARDING TIME COMPLEXITY COMPARISON WITH SEP

*Proof.* Suppose we have $h_{max}$ pooling layers in total, then the overall time complexity is $\mathcal{O}(\sum_{j=1}^{h_{max}} \sum_{i=1}^{n^{(j)}} d_{ij})$, $d_{ij}$ indicate the diameter for subgraph $i$ in $j$-th pooling layer. The worst case is that the input graph has no edge and get time complexity of $\mathcal{O}(n^{(k)})$. Thus we have overall time complexity $\mathcal{O}(\sum_{j=1}^{h_{max}} \sum_{i=1}^{n^{(j)}} d_{ij}) \leq \mathcal{O}(\sum_{j=1}^{h_{max}} n^{(j-1)}) \leq \mathcal{O}(h_{max} n^{(0)})$. Compared to SEP Wu et al. (2022) that also apply heuristic algorithm, our parsing algorithm has a better time efficiency: SEP has a overall time complexity $\mathcal{O}(2n^{(0)} + h_{max}(m^{(0)} \log n^{(0)} + n^{(0)}))$ (Wu et al., 2022), which is much larger than ours. □

## A.4  PROOFS REGRADING PROPOSITION 6

*Proof.* In order to prove the conclusion, we only need to demonstrate that any two nodes in the graph $G^{(k)} = (\mathcal{V}^{(k)}, \mathcal{E}^{(k)})$, if there is a path connect them, then a path would exist between the corresponding clusters in the graph $G^{(k+1)} = (\mathcal{V}^{(k+1)}, \mathcal{E}^{(k+1)})$.

Assuming nodes $i, j \in \mathcal{V}^{(k)}$, and there exists a path $s$ between them: $s = (i = v_0 - v_1 - \cdots - v_{N-1} - v_N = j)$, where $v_l \in \mathcal{V}^{(k)}, l = 0, \cdots, N$. Suppose the cluster assigned to node $v_l \in \mathcal{V}^{(k)}$ is $p_{v_l} \in \mathcal{V}^{(k+1)}$, now we examine the relationship between the clusters $p_{v_l}, p_{v_{l+1}}$. If the edge $(v_l, v_{l+1}) \in \mathcal{E}^{(k)}$ is a dominant edge, meaning $\hat{\mathbf{C}}^{(k)}_{v_l, v_{l+1}} > 0$. It should be noted that the operator $\mathrm{EXP}(\cdot)$ conducts a neighborhood-lookup operation on $\hat{\mathbf{C}}^{(k)}$, regardless of whether the node $v_l$ or

$v_{l+1}$ is traversed first in Stage 2 of the parsing algorithm, since these two nodes are adjacent in $\hat{\mathbf{C}}^{(k)}$, they will both be assigned to the same cluster, we have $p_{v_l} = p_{v_{l+1}}$.

If the edge $(v_l, v_{l+1}) \in \mathcal{E}^{(k)}$ is not a dominating edge, i.e., $\hat{\mathbf{C}}_{v_l,v_{l+1}}^{(k)} = 0$, then the two nodes will be assigned to different clusters. According to Equation 2, $\mathbf{A}^{(k+1)} = \mathbf{S}^{(k)^{\mathrm{T}}} \mathbf{A}^{(k)} \mathbf{S}^{(k)}$, then $\mathbf{A}_{p_{v_l},p_{v_{l+1}}}^{(k+1)} = (\mathbf{S}^{(k)^{\mathrm{T}}})_{p_{v_l},:} \mathbf{A}^{(k)} \mathbf{S}_{:,p_{v_{l+1}}}^{(k)} = \sum_{r=1}^{n^{(k)}} (\sum_{t=1}^{n^{(k)}} \mathbf{S}_{t,p_{v_l}}^{(k)} \mathbf{A}_{t,r}^{(k)}) \mathbf{S}_{r,p_{v_{l+1}}}^{(k)}$. Noting that $\mathbf{S}_{v_l,p_{v_l}}^{(k)} = \mathbf{S}_{v_{l+1},p_{v_{l+1}}}^{(k)} = 1, \mathbf{A}_{v_l,v_{l+1}}^{(k)} = 1$, when $t = v_l, r = v_{l+1}$, the corresponding summation term is 1, which means that at least one non-zero term exists among all the summed terms. Since all summation items are non-negative, it follows that $\mathbf{A}_{p_{v_l},p_{v_{l+1}}}^{(k+1)} > 0$. We have $(p_{v_l}, p_{v_{l+1}}) \in \mathcal{E}^{(k+1)}$.

In conclusion, $p_{v_l}$ and $p_{v_{l+1}}$ are either the same node or adjacent nodes. Therefore, the node sequence $(p_{v_0}, \cdots, p_{v_N})$ is a path on the graph $G^{(k+1)}$. This completes the proof. $\qquad\square$

## B  RELATED WORKS

**Graph pooling** Researchers have shifted their focus to hierarchical pooling as flat pooling methods (Atwood & Towsley, 2016; Xu et al., 2019) face difficulties in capturing comprehensive structural information. Early node dropping methods (Gao & Ji, 2019; Lee et al., 2019) only retain nodes with higher computed scores. This explicit dropping risks the irreversible loss of important node information and can break the graph connectivity (Baek et al., 2021; Wu et al., 2022). To better preserve node information, InfoMax-based (Li et al., 2020), memory-based (Khasahmadi et al., 2020) and global prototype-based (Lee et al., 2022) techniques have been suggested; Zhang et al. (2019; 2020); Ranjan et al. (2020) improves graph connectivity by graph structure learning. These attempts alleviate the problem of node dropping, but at cost of additional computing burden. On the other hand, the node clustering approach calculates a soft assignment vector for each node. This method effectively retains the node's information but can be memory-intensive due to its dense assignment matrix. Some works try to make this matrix sparser using various priors, such as penalty entropy in Ying et al., or Haar wavelet transforms in Wang et al.. However, these penalty terms may hurt model training. Another drawback of node clustering methods is that they can easily converge to trivial solutions where each node has uniform soft assignments, due to the ignorance of locality. Several methods inject locality into GNN encoders, such as CRF-based (Yuan & Ji, 2020) or MinCut-based (Bianchi et al., 2020), but they are unable to scale to larger graphs. Unlike the previous mentioned works, Jo et al. (2021) focuses on edges as the center and emphasizes precise learning of both node and edge information. Finally, none of these approaches can optimize the pooling structure for each individual graph.

**Structure Optimization** Structure optimization on graphs has gained significant attention in recent times. The majority of these methods concentrate on the *horizontal level*, which involves learning graph structure through techniques such as structure learning (Jiang et al., 2019; Chen et al., 2020b; Jin et al., 2020; Suresh et al., 2021) to determine node adjacency with or without a graph prior, or graph rewiring (Gasteiger et al., 2019; Topping et al., 2022; Bo et al., 2023) that adds or removes edges to enhance message passing. Only a few focus on the *vertical level*, which is concerned with learning pooling structures. EdgePool (Diehl, 2019) is an initial attempt to dynamically learn pooling trees by iteratively contracting edges based on calculated scores, unless one of their nodes has already been part of a contracted edge. Although they achieve end-to-end pooling tree learning through edge scores, their contraction-based strategy limits flexibility as it results in a near-half pooling ratio. On the other hand, SEP (Wu et al., 2022) uses combinatorial optimization to generate pooling trees by defining and optimizing a metric with a greedy algorithm. While this optimizes the pooling tree well for each graph, it is only used as a pre-processing step before training and cannot benefit from downstream supervision signals. Additionally, SEP relies solely on graph topology and fails to capture node features.

**Grammar Induction** Grammar induction, which aims at learning the latent structure behind natural language in an unsupervised way, can be dated back to Klein & Manning (2004). It has seen many advances recently, such as by using amortized variational inference on a transition-based parser Kim et al. (2019b), by dynamic programming Drozdov et al. (2019), or by maximizing the marginal likelihood of a sentence generated by PCFG Kim et al. (2019a), etc. Among all these various approaches, the most related is the works based on *syntactic distance* Shen et al. (2018b), which

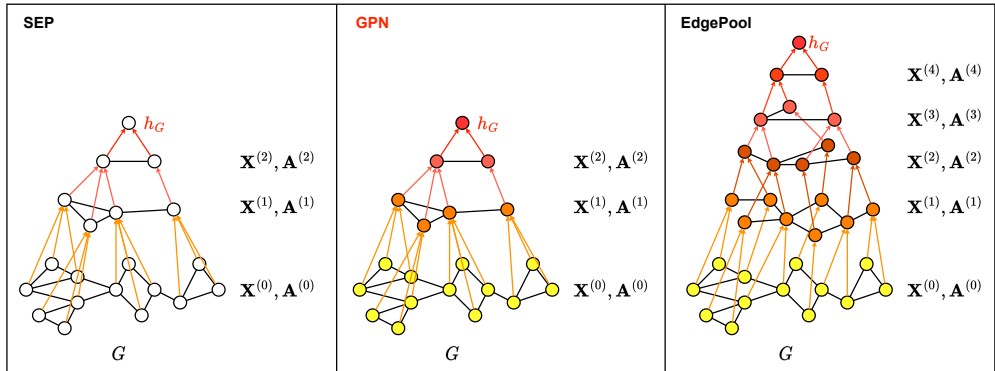

Figure 8: A comparison for SEP, EdgePool and GPN in terms of their pooling mechanisms.

is a sequence of continuous scalar values that allows the tree to be inferred from it in a bottom-up fashion. This approach has been successfully shown to work for both recurrent models Shen et al. (2018a; 2019) and Transformer-based models Shen et al. (2021). Different from the above works that induce latent tree structures as a byproduct of some self-supervised training process, our approach let the latent structure drive graph pooling and engage in gradient backpropagation, resulting in a more task-relevant model.

## C    DIFFERENCES COMPARED TO CLOSE METHODS

Our model significantly differs from EdgePool. In EdgePool, each cluster is usually composed of two adjacent nodes, and only when all neighbors of a certain node have been assigned, a cluster with a single node will be formed. Therefore, the proportion of pooling in the EdgePool is always close to 50%, which is not learnable. In contrast, in our model, the pooling ratio is not constrained and is entirely inferred by the parsing algorithm, allowing for end-to-end learning. Moreover, the number of pooling layers in EdgePool is set as a hyperparameter, and each pooling layer uses independent parameters. However, we recursively use a single pooling layer to construct our model, the number of which is inferred by the parsing algorithm. This varies continuously during the training process, with all pooling layers sharing the same parameters. Figure 8 visually illustrates the differences between three methods for optimizing pooling tree, i.e., SEP, EdgePool, and our proposed GPN.

## D    LIMITATION

In Section 3.4, our model demonstrates good time efficiency when processing batched graphs. However, if a batch contains graphs with varying numbers of pooling layers, the model may not be as time-efficient. This is because graphs with fewer pooling layers would have to wait for those with more iterations to finish before proceeding, potentially leading to inefficiencies when dealing with pretty large batches of graph data. For potential extreme cases that the inferred height of pooling tree is too large, we can set a maximum height limit for the pooling tree in the algorithm. This means the algorithm will stop during the parsing process once this height is reached.

## E    DISCUSSION

When running our model on a large-scale graph with a significant number of nodes, our graph parsing algorithm may infer a high pooling tree. In this case, the model will perform numerous graph convolutions and continuously smooth node features. The model structure for node classification task already includes residual connections from the encoder to the decoder, which helps alleviate potential over-smoothing issues by preserving local neighborhood information and original node features through residual connections.

## F    DETAILS FOR EQUATION 4 AND END-TO-END TRAINING

Here we describe the forward and backward passes of our model to clarify the training process: During the forward pass, we first use GNN and MLP to compute the edge scores (Equation 1). Then we run Algorithm 1 to obtain the assignment matrix (Equation 2) and performs the multiset computation to obtain the coarsened graph (Equation 3). The final step is to compute the edge score mask (Equation 4) to ensure the gradients can flow into GNN and MLP. During the backward pass, Algorithm 1 does not run. Instead, the gradient backpropagation is performed directly. The GNN and MLP can be updated through the gradients of the edge score mask.

We further present the details of Equation 4 to enhance understanding: Induce subgraph $\hat{G}_i^{(k)} \subseteq G^{(k)}$ from node $v_i$ means: each node $v_i$ in graph $G^{(k+1)}$ is a cluster generated by the $k$-th pooling layer, thus it corresponds to a connected subgraph $\hat{G}_i^{(k)}$ in $G^{(k)}$. In order to allow gradients to flow into the MLP used for calculating edge scores, we aggregate the scores of all the edges $\hat{\mathcal{E}}_i^{(k)}$ in each subgraph $\hat{G}_i^{(k)}$. We then obtain a vector $\mathbf{y}_i^{(k)} \in \mathbb{R}^{n^{(k+1)}}$ represents the score of clusters. This vector is then replicated into a mask matrix $\mathbf{y}^{(k)}\mathbf{1}^\mathrm{T} \in \mathbb{R}^{n^{(k+1)} \times d}$, which matches the shape of the node feature matrix $\hat{\mathbf{X}}^{(k+1)}$. Finally, this mask is element-wise multiplied with the node feature matrix. During the backward pass, Algorithm 1 does not run. Instead, the gradient backpropagation is performed directly. The GNN and MLP can be updated through the gradients of the edge score mask.

## G    BROADER IMPACTS

In this study, we suggest a new graph pooling model that can address issues with current methods. Our model derives the pooling structure from both the topology of the graph and the node features, and is not founded on any assumptions regarding the data, which mitigates the risk of social bias. Nonetheless, since the algorithm used for data parsing is data-driven, it is possible that the pooling structure may be misguided by corrupted graph structure or features, which could lead to unfavorable social consequences. Since both internet data and social networks can be presented as graphs, we need to establish ethical guidelines and limit the usage of our model to avert such issues.

## H    EXPERIMENTAL DETAILS

We use NVIDIA GeForce RTX 3090 and NVIDIA GeForce RTX 4090 as the hardware environment, and use PyTorch and PyTorch Geometric (Fey & Lenssen, 2019) as our software environment. The datasets are downloaded from PyTorch Geometric, we also use dataloader provided by PyTorch Geometric to load and pre-process datasets. We fix the random seed for reproducibility. We report detailed configurations as follows.

### H.1    GRAPH CLASSIFICATION

**Dataset details**    DD and PROTEINS are datasets of protein structures that represent macromolecules. Our task is to distinguish whether they are enzymes or non-enzymes based on their structure. The graph size of DD is larger than the other four datasets, resulting in the highest number of graph-pooling layers. NCI1 and NCI109 are two medium-sized datasets of chemical compounds labeled according to whether they are active against lung cancer cells and/or ovarian cancer cells. FRANKENSTEIN is a chimeric dataset of Mutagenicity and MNIST. It is relatively small and has the lowest average number of nodes among these five datasets, as reflected in its smallest number of learned graph-pooling layers. ogbg-molpcba is a chemical compound dataset used to determine molecular properties. It is a curated dataset from PubChem BioAssay (PCBA) that provides information on the biological activities of small molecules. The performance of ogbg-molpcba is measured using Average Precision, as this dataset is highly biased due to the small number of positive examples. We borrowed the baseline results from Lee et al. (2019), Ranjan et al. (2020), and Lee et al. (2022). Except for MinCutPool, HoscPool, GMT and SEP, we reran their code and report the performance, since they did not have public records under our setting. We followed PyTorch Geometric (Fey & Lenssen, 2019) scripts to reimplement EdgePool for testing.

Table 5: Best hyper-parameters for graph classification

| Dataset | Hyper-parameters |
|---------|------------------|
| DD | $L_{GNN} : 2, L_{DeepSets} : 1, Dropout : 0.2, lr : 0.0001, H : 128, B : 64$ |
| PROTEINS | $L_{GNN} : 3, L_{DeepSets} : 1, Dropout : 0.1, lr : 0.0005, H : 128, B : 128$ |
| NCI1 | $L_{GNN} : 2, L_{DeepSets} : 3, Dropout : 0.4, lr : 0.0005, H : 256, B : 256$ |
| NCI109 | $L_{GNN} : 3, L_{DeepSets} : 1, Dropout : 0.2, lr : 0.0005, H : 128, B : 64$ |
| FRANKENSTEIN | $L_{GNN} : 3, L_{DeepSets} : 1, Dropout : 0.2, lr : 0.001, H : 128, B : 128$ |
| ogbg-molpcba | $L_{GNN} : 3, L_{DeepSets} : 1, Dropout : 0.3, lr : 0.001, H : 128, B : 128$ |

Table 6: Best hyper-parameters for node classification

| Dataset | Hyper-parameters |
|---------|------------------|
| Cora | $L_{GNN_1} : 1, L_{GNN_2} : 2, L_{DeepSets} : 2, L_{MLP} : 1, Dropout : 0.5, DropEdge : 0.5, lr : 0.005$ |
| CiteSeer | $L_{GNN_1} : 2, L_{GNN_2} : 3, L_{DeepSets} : 2, L_{MLP} : 3, Dropout : 0.6, DropEdge : 0.1, lr : 0.005$ |
| PubMed | $L_{GNN_1} : 2, L_{GNN_2} : 2, L_{DeepSets} : 1, L_{MLP} : 1, Dropout : 0.7, DropEdge : 0.7, lr : 0.001$ |
| Film | $L_{GNN_1} : 2, L_{GNN_2} : 2, L_{DeepSets} : 1, L_{MLP} : 2, Dropout : 0.5, DropEdge : 0.5, lr : 0.001$ |

**Implementation details** For each experiment, we repeat the experiment on 20 seeds (from 0 to 19), and we use the 10-fold cross-validation method. So for each dataset we end up reporting the average result of 200 runs on the test set, which allows us to effectively avoid noise in the results and makes the results more credible. For all datasets, we use an early stopping strategy to avoid overfitting. We use validation loss as the criterion the early stopping. Also, the maximum number of epochs is set to 500 for all datasets. The Table 5 gives the best hyperparameters used in the 5 graph classification datasets. For the sake of brevity, we use abbreviations in the table. $L_{GNN}$ represents number of GCN layers inside the GNN block, $L_{DeepSets}$ represents number of MLP layers of DeepSets, $L_{MLP}$ represents number of layers of the MLP for edge score computation, $Dropout$ represents dropout in the MLP, $DropEdge$ represents the ratio of the edge dropout when perform parsing, $lr$ represents learning rate, $H$ represents hidden channel, $B$ represents batch size.

## H.2 NODE CLASSIFICATION

**Dataset details** CiteSeer, Cora and PubMed are all citation network datasets. They share a common feature in that their node features are 0/1-valued word vectors. CiteSeer consists of 3,312 scientific publications, which contain a total of six categories. It has a citation network consisting of 4,732 links. Its word vector has a dimension of 3,703. Cora consists of 2,708 scientific publications containing a total of seven categories. Its citation network consists of 5,429 links. Its word vector has a dimension of 1,433. PubMed consists of 19,717 scientific publications related to diabetes, which are divided into three different categories. Its citation network consists of 44,338 links. Its word vector has a dimension of 500. We obtain all baseline results from Zhu et al. (2020), Yan et al. (2022) and Bodnar et al. (2022), except for TopKPool and SEP, we rerun their code and perform a grid search on Cora, CiteSeer, and PubMed to ensure a fair comparison.

**Implementation details** For the node classification task, we used a ten-fold cross-validation method. We keep the same hidden layer for all datasets and limit the maximum number of epochs to 2000. we use an early stopping strategy for all datasets. We also use validation loss as an early stopping criterion. We use the grid search method to find suitable hyperparameters. Table 6 gives the different hyperparameters used for the four node classification datasets. And unlike the first three datasets, Film is a dataset that describes the collaborative relationships between actors.

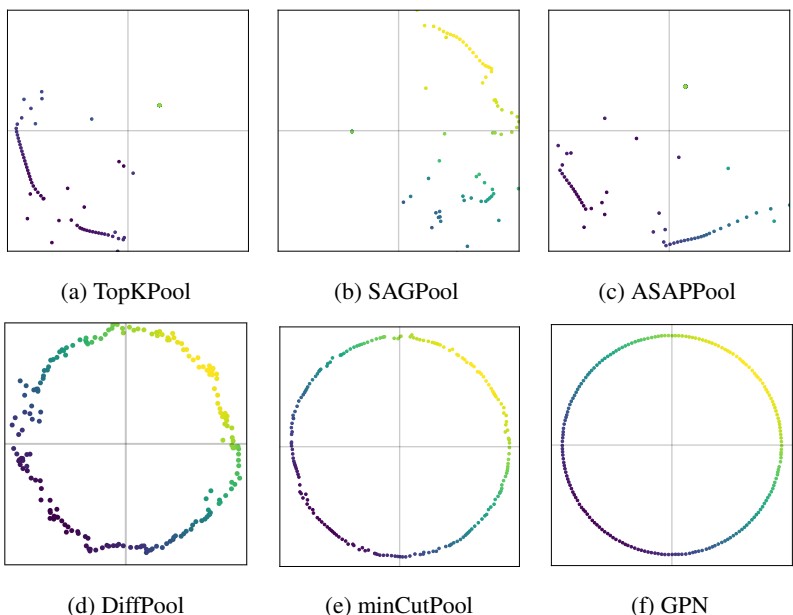

(a) TopKPool        (b) SAGPool        (c) ASAPPool

(d) DiffPool        (e) minCutPool        (f) GPN

Figure 9: Reconstruction results of ring synthetic graphs, compared to node drop and clustering methods.

## H.3 GRAPH RECONSTRUCTION

**Implementation details** In order to keep the same setting with Bianchi et al. (2020), we use two message passing layers before the pooling operation and after the unpooling operation. We choose mean squared error (MSE) as the loss function. We use an early stopping strategy where patience is set to 1,000 epochs. The maximum number of epochs is set to 10,000. We then optimise the network using the Adam optimiser. In the experiment of Bianchi et al. (2020), they retained 25% of the nodes by adjusting the height of the tree. In contrast, our GPN model automatically retains around 30% of the nodes of the grid and ring during pooling.

**More reconstruction results** Figure 9 and Figure 10 show the results of graphs reconstructed using various pooling methods. We can visually see that the node-drop method suffers from information loss, resulting in the original graph being basically unrecognisable. Also, we can see that GPN preserves the shape of the ring as well as the coordinates of the interior of the mesh very well.

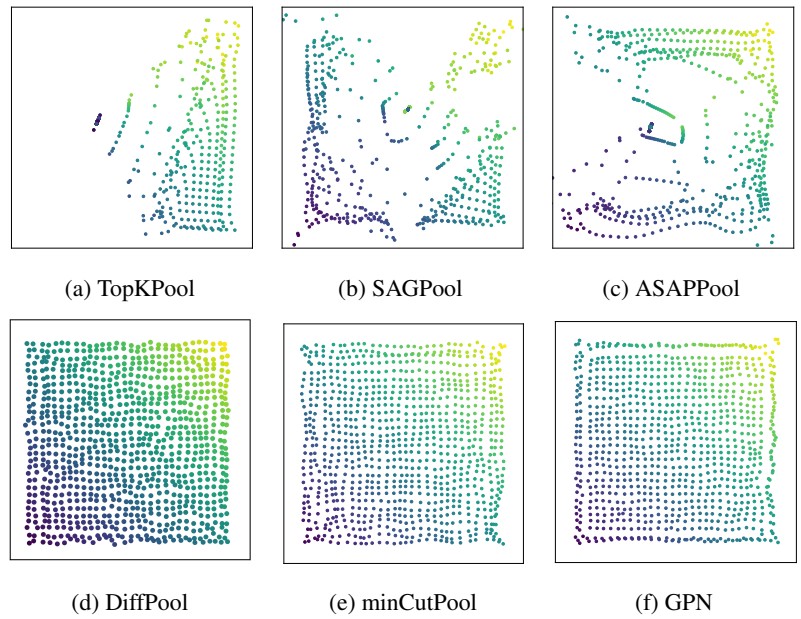

(a) TopKPool     (b) SAGPool     (c) ASAPPool

(d) DiffPool     (e) minCutPool     (f) GPN

Figure 10: Reconstruction results of grid synthetic graphs, compared to node drop and clustering methods.

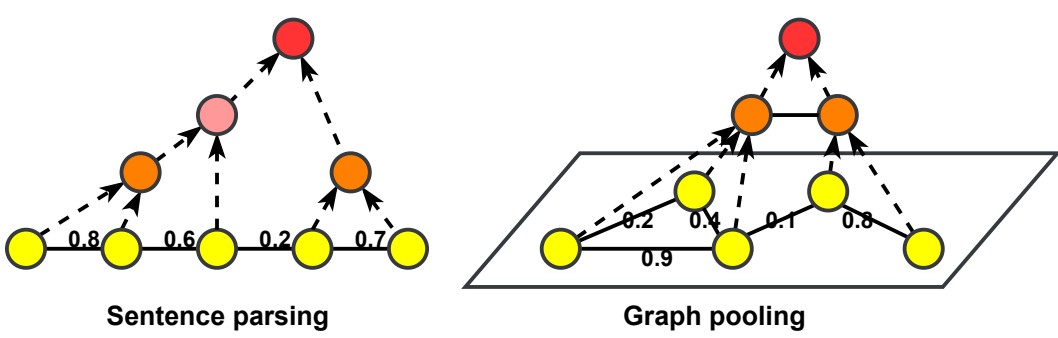

Figure 11: Sentence parsing build a syntactic tree based on the scores between tokens; graph parsing build a pooling structure based on the scores defined on edges.

