# OpenReview forum: "Graph Parsing Networks"
_ICLR.cc/2024/Conference — ICLR 2024 poster_

### Official Review · Reviewer_UKss · 2023-10-26

**Soundness:** 2 fair
**Presentation:** 3 good
**Contribution:** 3 good
**Rating:** 5
**Confidence:** 3

**Summary:**

This work proposes an efficient graph parsing algorithm to infer the pooling structure that drives graph pooling. The pooling structure is trained end-to-end. It achieves competitive real-world performance and good memory and time efficiency.

**Strengths:**

1. Clear presentation of method.

2. Extensive experiments on both graph and node tasks.

**Weaknesses:**

1. Time complexity. Though the paragraph above Proposition 4 claims that the time complexity of each pooling layer is $O(n^{(k)})$, the computation of edge score $C$ still takes $O(m^{(k)})$ time.

2. Proposition 5 is doubtful. The argmax operator may not produce unique output when the multiset has several equal maximum elements.

3. The performance gain in ablation study is not significant. In Table 4, the performance difference between the origin model and the model with  Mean pool is smaller than the score deviation.

**Questions:**

1. The point 1 and 2 in the Weakness section.

2. The argmax operator in DOM is not differentiable. How to train the whole model end-to-end?

---

> ### Author Response · Authors · 2023-11-17
> **Response to Reviewer UKss**
>
> Thank you for the thoughtful feedback on our manuscript. We provide the following detailed responses to your major concerns.
>
> > (Weakness 1) Time complexity. Though the paragraph above Proposition 4 claims that the time complexity of each pooling layer is $O(n^{(k)})$, the computation of edge score $C$ still takes $O(m^{(k)})$ time.
>
> The computation of the edge score $C$ indeed has complexity $\mathcal{O}(m^{(k)})$. However, it occurs before Algorithm 1, and in the paragraph above Proposition 4, we specifically focused on the time complexity of Algorithm 1 as $\mathcal{O}(n^{(k)})$. The computation of $C$ is dominated by message passing, which can be computed in parallel on GPU. On the other hand, the parsing algorithm must be executed sequentially. That's why, in our time efficiency tests, even EdgePool with $\mathcal{O}(m^{(k)})$ complexity runs very slowly when dealing with densely connected graphs.
>
> > (Weakness 2) Proposition 5 is doubtful. The argmax operator may not produce unique output when the multiset has several equal maximum elements.
>
> Thank you for pointing this out! Yes, if there are two nodes with common neighbors and their node features are exactly the same, then the output of the argmax operator cannot guarantee uniqueness. Based on your suggestion, we edited Proposition 5 and it's corresponding proof (paragraph 2 in Appendix A.2) in the updated submission, please refer to the red-colored text. Specifically, we suppose the row-wise non-zero entries in $C$ are distinct. In practice, we find that the potential violation of the permutation invariance caused by the argmax operator has a limited impact on model performance.
>
> > (Weakness 3) The performance gain in ablation study is not significant. In Table 4, the performance difference between the origin model and the model with Mean pool is smaller than the score deviation.
>
> In our ablation study, we aim to verify that the performance improvement is primarily due to the parsing algorithm we proposed and to show that our model is robust to changes in the backbone. After replacing the DeepSets module with a simpler Mean pool, the model continues to deliver strong performance, confirming its robustness to the choice of backbone and supporting our claim.
>
> ---
>
> > (Question) The argmax operator in DOM is not differentiable. How to train the whole model end-to-end?
>
> Here we describe the forward and backward passes of our model to clarify the training process:
> - During the forward pass, we first use GNN and MLP to compute the edge scores (Equation 1). Then we run Algorithm 1 to obtain the assignment matrix (Equation 2) and performs the multiset computation to obtain the coarsened graph (Equation 3). The final step is to compute the edge score mask (Equation 4) to ensure the gradients can flow into GNN and MLP.
> - Here, we further present the details of Equation 4 to enhance understanding:
>     - Induce subgraph $\hat{G}_i^{(k)} \subseteq G^{(k)}$ from node $v_i$ means: each node $v_i$ in graph $G^{(k+1)}$ is a cluster generated by the $k$-th pooling layer, thus it corresponds to a connected subgraph $\hat{G}_i^{(k)}$ in $G^{(k)}$.
>     - In order to allow gradients to flow into the MLP used for calculating edge scores, we aggregate the scores of all the edges $\hat{\mathcal{E}}\_i^{(k)}$ in each subgraph $\hat{G}\_i^{(k)}$. We then obtain a vector $\mathbf{y}^{(k)}\_{i} \in \mathbb{R}^{n^{(k+1)}}$ represents the score of clusters.
>     - This vector is then replicated into a mask matrix $\mathbf{y}^{(k)} \mathbf{1}^{\mathrm{T}} \in \mathbb{R}^{n^{(k+1)} \times d}$, which matches the shape of the node feature matrix $\mathbf{\hat{X}}^{(k+1)}$. Finally, this mask is element-wise multiplied with the node feature matrix.
> - During the backward pass, Algorithm 1 does not run. Instead, the gradient backpropagation is performed directly. The GNN and MLP can be updated through the gradients of the edge score mask.

---

> > ### Comment · Reviewer_H5gS · 2023-11-21
> >
> > I am happy with the revisions made. Thanks to the authors for their efforts.

---

> > > ### Author Response · Authors · 2023-11-22
> > > **Thank you for your positive feedback**
> > >
> > > Dear Reviewer H5gS, thank you for your positive feedback on the revisions made to our paper. We would like to receive any specific suggestions or feedback that could help further improve the paper and potentially raise the score. We are dedicated to improving the quality of our submission and would greatly appreciate your suggestions in this process.

---

> > ### Comment · Reviewer_UKss · 2023-11-22
> >
> > Thank you for the detailed response. However, I think the assumption that the row-wise non-zero entries in $C$ are distinct is too strong for permutation invariance in Proposition 5. When the graph is symmetric, equal entries are very likely to exist. I would still keep my score to 5.

---

> > > ### Author Response · Authors · 2023-11-22
> > > **Response to Reviewer UKss**
> > >
> > > Dear Reviewer UKss, thank you for your response. We would like to kindly clarify that there may be some misunderstanding regarding our assumptions, and we are more than happy to provide further clarification on this matter.
> > >
> > > - For the sake of brevity, we have shortened the text of the assumptions made, so you may have misunderstood the meaning of "row-wise non-zero entries". What we mean is that for any given row $i$ in matrix $C$, all non-zero entries of $C_i$ are distinct from each other, rather than stating that the non-zero entries of any $C_i$ and $C_j$ are mutually exclusive multisets. In other words, our assumption is that for each node $i$, the edge scores $\{ C_{ij} | j \in \mathcal{N}(i) \}$ are distinct. According to our computation method for edge scores (see Equation 1), this assumption is only broken when the two neighbors $j,k$ of the central node $i$ have exactly the same node features (as we explained in *Response to Weakness 2* and as stated in Appendix A.2, paragraph 2). Therefore, we consider this assumption not to be strong, which is also reflected in the model's good performance.
> > >
> > > - Thus for the example of a symmetric graph you mentioned, assuming nodes $i,j$ are a pair of nodes symmetrically positioned in the graph, even if the multisets of non-zero entries in $C_i$ and $C_j$ are equal, as long as the non-zero entries within $C_i$ and $C_j$ are distinct, the permutation invariance can still be guaranteed.
> > >
> > > We sincerely apologize for any confusion that may have arisen and hope that our explanation can offer you a clearer understanding of the assumptions presented in Proposition 5. If you have any lingering uncertainties, we are more than willing to provide further clarification and address your concerns.

---

### Official Review · Reviewer_xTBe · 2023-10-30

**Soundness:** 3 good
**Presentation:** 4 excellent
**Contribution:** 3 good
**Rating:** 8
**Confidence:** 3

**Summary:**

This paper aims to construct a graph pooling method called a graph parsing network (GPN) that can be personalized to every graph and has a good balance of maintaining the node information and high memory efficiency simultaneously.
The method is built upon existing GNNs and designs a novel algorithm to construct personalized pooling trees for graphs. The empirical results on graph classification show its effectiveness and efficiency.

**Strengths:**

1. The paper is well written, clearly showing its motivation and model structure.
2. The novel part lies in that it designs a novel algorithm that can adaptively form clusters layer by layer that minimizes loss of information of graph structure and node features, and since it does not have learnable parameters in the graph parsing algorithm.
3. The empirical results on graph classification and reconstruction demonstrate that the proposed model has a good ability to maintain the graph structure and node feature information.

**Weaknesses:**

1. It seems that GPN works well on small-sized graphs from the experimental results. Discussion on large-sized graphs is expected to be included in the paper.
2. If the tree is high for some large-sized graph, the performance might be worse due to over-smoothing.
3. For the model itself involving GNNs, it is not clear how you train this model.

**Questions:**

1. What does a 'different GNN` mean in node classification? $\mathbf{H}^{(k)}$ in Eq. 3 is the same as '$GNN^\prime(\mathbf{X}^{(k)}, \mathbf{A}^{(k)})$` in eq. 10.

---

> ### Author Response · Authors · 2023-11-17
> **Response to Reviewer xTBe**
>
> We sincerely appreciate the reviewer's constructive feedback and positive remarks on our work. We provide the following detailed responses to your major concerns.
>
> > (Weakness 1) It seems that GPN works well on small-sized graphs from the experimental results. Discussion on large-sized graphs is expected to be included in the paper.
>
> > (Weakness 2) If the tree is high for some large-sized graph, the performance might be worse due to over-smoothing.
>
> It is worth noting that, the PubMed dataset we tested already contains 18,717 nodes, which is not that small-sized. Following your suggestion, in the updated submission (please refer to the red-colored text), we further strengthen the discussion on large graphs in Appendix E. In short, our node classification model architecture already includes residual connections from the encoder to the decoder (see Section 2.3 and Figure 3), which can help mitigate potential over-smoothing issues on large graphs by preserving local information.
>
> > (Weakness 3) For the model itself involving GNNs, it is not clear how you train this model.
>
> Since the Reviewer UKss raised the same question, we kindly request that you refer to *Response to Reviewer UKss - Question*.
>
> ---
>
> > (Question) What does a different GNN mean in node classification? $\mathbf{H}^{(k)}$ in Eq. 3 is the same as $GNN^\prime(\mathbf{X}^{(k)}, \mathbf{A}^{(k)})$ in eq. 10.
>
> In Equation 3, the multiset computation module directly use the node feature $\mathbf{H}^{(k)}$, which is encoded by the GNN in Equation 1. While in Equation 10, the input node feature is $\mathbf{X}^{(k)}$, and we use another GNN' (different from the GNN in Equation 1) to encode it.

---

### Official Review · Reviewer_4RVW · 2023-10-31

**Soundness:** 3 good
**Presentation:** 3 good
**Contribution:** 3 good
**Rating:** 8
**Confidence:** 5

**Summary:**

This paper presents a novel graph pooling method, namely Graph Parsing Network (GPN), which aims to learn an individual pooling structure for each individual graph in an end-to-end fashion while being memory and time efficient. Specifically, the authors generate the assignment matrix, which is used to transform the larger graph into its compressed smaller graph by mapping nodes of the original graph into certain clusters, per graph. Also, this assignment matrix is generated based on the graph parsing algorithm that infers pooling structures of graphs from their edge scores, i.e., firstly selecting the dominant edges and then expanding them to construct the node assignment matrix for graph pooling. The authors evaluate the performance of the proposed Graph Parsing Network (GPN) on graph classification and node classification tasks as well as the graph reconstruction task, showing its effectiveness and efficiencies.

**Strengths:**

* The motivation to adaptively condense different graphs based on their differences in graph structures (i.e., edge importances) is sound.
* The idea of using a graph parsing algorithm to make an individual pooling structure for each graph is interesting and reasonable.
* The proposed GPN surpasses existing graph pooling methods on both graph-level and node-level tasks.
* The proposed GPN is efficient in terms of memory usage and computational time, particularly compared to the node-clustering-based (assignment-matrix-based) graph pooling methods.

**Weaknesses:**

* This work may be similar to the idea of [1] which considers and manipulates edges during graph pooling (i.e., making the cluster assignment matrix with edges), since this work also performs edge-centric graph pooling with the cluster assignment matrix that is generated from edges.
* Equation (4) lacks details on motivation. I understand that, in order to perform backpropagation with edge scores, edge scores should be added or multiplied with other learnable variables. On the other hand, it is unclear why the edge scores should be multiplied in this way (Equation (4)) and what may be the effect of this multiplication on node features.
* The authors may further perform the statistical test on the main results (Table 1 and Table 2) since the performance of the proposed GPN is comparable against other performant baselines when considering mean and standard deviations together.

---

[1] Edge Representation Learning with Hypergraphs, NeurIPS 2021.

**Questions:**

* In Equation (10), it may be better to represent the superscript on GNN' according to its layer index, since GNN' is different across different layers.
* In Figure 5, the proposed GPN is memory efficient compared to the baseline that uses the cluster assignment matrix for graph pooling. However, the GPN also uses the cluster assignment matrix for graph pooling. In this vein, I am wondering why there exists a significant difference in efficiency while both methods are similar in using the cluster assignment matrix.

---

> ### Author Response · Authors · 2023-11-17
> **Response to Reviewer 4RVW**
>
> We sincerely appreciate the reviewer's considerate feedback. Here, we provide detailed responses to your concerns.
>
> > (Weakness 1) This work may be similar to the idea of [1] which considers and manipulates edges during graph pooling (i.e., making the cluster assignment matrix with edges), since this work also performs edge-centric graph pooling with the cluster assignment matrix that is generated from edges.
>
> Thank you for mentioning this work. In our updated submission (please refer to the red-colored text), we have introduced it in Introduction (paragraph 3 in Section 1) and Related Works (paragraph 1 in Appendix B).
>
> > (Weakness 2) Equation (4) lacks details on motivation. I understand that, in order to perform backpropagation with edge scores, edge scores should be added or multiplied with other learnable variables. On the other hand, it is unclear why the edge scores should be multiplied in this way (Equation (4)) and what may be the effect of this multiplication on node features.
>
> - Yes, the multiplication is for backpropagation, this approach is similar to [1] (see the last equation in Equation 2).
>
> - In this way (adding the score and then multiplying), the output cluster from graph pooling can be aware of the number of edges assigned to it through its features.
>
> - Here, we present the details of Equation 4 to enhance understanding:
>     - Induce subgraph $\hat{G}_i^{(k)} \subseteq G^{(k)}$ from node $v_i$ means: each node $v_i$ in graph $G^{(k+1)}$ is a cluster generated by the $k$-th pooling layer, thus it corresponds to a connected subgraph $\hat{G}_i^{(k)}$ in $G^{(k)}$.
>     - In order to allow gradients to flow into the MLP used for calculating edge scores, we aggregate the scores of all the edges $\hat{\mathcal{E}}\_i^{(k)}$ in each subgraph $\hat{G}\_i^{(k)}$. We then obtain a vector $\mathbf{y}^{(k)}\_{i} \in \mathbb{R}^{n^{(k+1)}}$ represents the score of clusters.
>     - This vector is then replicated into a mask matrix $\mathbf{y}^{(k)} \mathbf{1}^{\mathrm{T}} \in \mathbb{R}^{n^{(k+1)} \times d}$, which matches the shape of the node feature matrix $\mathbf{\hat{X}}^{(k+1)}$. Finally, this mask is element-wise multiplied with the node feature matrix.
>
> > (Weakness 3) The authors may further perform the statistical test on the main results (Table 1 and Table 2) since the performance of the proposed GPN is comparable against other performant baselines when considering mean and standard deviations together.
>
> Following you suggestion, we performed t-tests and report the best model and its comparable models (p > 0.05) here:
>
> - Table 1:
>     | **Dataset**   | **Best model** | **Comparable models**                  |
>     |---------------|----------------|-------------------------------------|
>     | **NCI1**      | **GPN**            | -                                   |
>     | **NCI109**    | **GPN**            | -                                   |
>     | **FRANKENSTEIN** | **GPN**          | EdgePool                            |
>     | **PROTEINS**  | **GPN**            | GMT, SPGP, MinCutPool, HoscPool, SEP |
>     | **DD**        | GMT            | SPGP                                |
> - Table 2:
>     | **Dataset** | **Best model** | **Comparable models**             |
>     |-------------|----------------|--------------------------------|
>     | **Film**    | **GPN**            | GCNII, GGCN, NSD               |
>     | **Cora**    | GCNII          | **GPN**, MixHop, H2GCN, GPRGNN, GGCN, NSD |
>     | **CiteSeer**| Geom-GCN       | **GPN**, GCNII, H2GCN, GPRGNN, GGCN, NSD |
>     | **PubMed**  | GCNII          | Geom-GCN                       |
>
> ---
>
> > (Question 1) In Equation (10), it may be better to represent the superscript on GNN' according to its layer index, since GNN' is different across different layers.
>
> The GNN' in Equation 10 is shared across the pooling layers, thus we omitted the superscript.
>
> > (Question 2) In Figure 5, the proposed GPN is memory efficient compared to the baseline that uses the cluster assignment matrix for graph pooling. However, the GPN also uses the cluster assignment matrix for graph pooling. In this vein, I am wondering why there exists a significant difference in efficiency while both methods are similar in using the cluster assignment matrix.
>
> This is because the clustering-based methods need to store a dense assignment matrix with quadratic memory complexity (see Section 1 and Figure 1), while our model ensures linear memory complexity through the graph parsing algorithm (see paragraph 5 in Section 2.2).
>
> ---
>
> [1] Graph U-Nets, ICML 2019

---

> > ### Comment · Reviewer_4RVW · 2023-11-18
> >
> > Thank you for responding to my comments and addressing them. Regarding Weakness 2, I suggest authors more clearly explain the process and reason (of Equation (4)) for multiplying edge scores with the node features in the paper. Besides this, I do not have any additional major concerns.

---

> > > ### Author Response · Authors · 2023-11-18
> > > **Response to Reviewer 4RVW**
> > >
> > > Thank you for your suggestion! We have included the details and reasons for Equation 4 in the updated submission, highlighted with red-colored text. Please refer to paragraph 5 in Section 2.1 and Appendix F.

---

> > > > ### Comment · Reviewer_4RVW · 2023-11-19
> > > >
> > > > Thank you for your response. The update looks good to me. As the authors address all of my comments and reflect them in the paper, as well as I believe there are no further major concerns (after reading other reviews as well), I have increased the score to Accept.

---

> > > > > ### Author Response · Authors · 2023-11-19
> > > > > **Thank you for your positive feedback**
> > > > >
> > > > > Many thanks for your positive remarks. Thank you for the opportunity to enhance the quality of our research.

---

### Official Review · Reviewer_H5gS · 2023-11-04

**Soundness:** 2 fair
**Presentation:** 2 fair
**Contribution:** 2 fair
**Rating:** 3
**Confidence:** 4

**Summary:**

This paper deals with graph pooling, i.e., compressing graph information into compact representations. The paper contains a mostly experimental study of various methods. The proposed graph parsing network constitutes a novel contribution, it adaptively learns personalized pooling structures. Experimental results show that the method outperforms state of the art graph pooling methods.

**Strengths:**

The proposed approach does seem novel, and it also seems to perform well compared to existing graph pooling networks.

**Weaknesses:**

Graph pooling can elegantly be cast into a compression framework and would hence benefit from an analysis of
its fundamental limits based on information theory. The pooling of a graph structure (potentially with some side information as
considered) here is a classical information-theoretic problem, i.e., how to best store all the information characterizing the graph
in a vector of finite length or even better a bitstring of finite length. It may well be that the entire field of graph pooling as
currently considered in the ML literature does not take this perspective, but reading this paper makes it clear that many of the questions
asked here would benefit from such a perspective.

Generally, this reviewer finds the paper to be quite ad-hoc and also it uses a lot of jargon, not always defined or consistently used. The language is also confusing in many places.

**Questions:**

none

---

> ### Author Response · Authors · 2023-11-17
> **Response to Reviewer H5gS**
>
> Thank you for your comments. Here is our response.
>
> > (Weakness 1) Graph pooling can elegantly be cast into a compression framework and would hence benefit from an analysis of its fundamental limits based on information theory. The pooling of a graph structure (potentially with some side information as considered) here is a classical information-theoretic problem, i.e., how to best store all the information characterizing the graph in a vector of finite length or even better a bitstring of finite length. It may well be that the entire field of graph pooling as currently considered in the ML literature does not take this perspective, but reading this paper makes it clear that many of the questions asked here would benefit from such a perspective.
>
> We didn't mention the "information theory" in our paper, and the graph pooling problem we studied is not close to it.
>
> > (Weakness 2) Generally, this reviewer finds the paper to be quite ad-hoc and also it uses a lot of jargon, not always defined or consistently used. The language is also confusing in many places.
>
> Our paper is not ad-hoc. We didn't use jargon in our paper, and we have defined the symbols we used. We didn't use confusing language in our paper.

---

### Author Response · Authors · 2023-11-17
**Source code release**

You can view the source code, splitting files, hyper-parameters and experimental steps required to reproduce the results of this paper at the following anonymous link: https://anonymous.4open.science/r/GPN_ICLR24_SourceCode-07A3.

---

### Author Response · Authors · 2023-11-20
**General comment**

We thank all the reviewers for their time and comments. We updated our submission and included the following main changes (please refer to the red-colored text):

- Added details about Equation 4 and end-to-end training. (in Section 2.1 and Appendix F)
- Added reference and introduction to work [1]. (in Section 1 and Appendix B)
- Edited Proposition 5 and its proof, considering the situation where the argmax operator encounters multiple identical maximum values. (in Proposition 5 and Appendix A.2)
- Added a discussion on large graphs and their potential over-smoothing problem. (in Appendix E)

We hope our reply has addressed your concerns. We would appreciate to know if you have any additional questions, concerns or clarifications you would like to ask us.

---

[1] Edge Representation Learning with Hypergraphs, NeurIPS 2021

---

### Meta-Review · Area_Chair_KLFy · 2023-12-15

**Metareview:**

The paper addresses the challenges in graph pooling, aiming to compress graph information into a compact representation while adapting to individual graph structures. Existing methods often follow a hierarchical approach with fixed pooling ratios or layer numbers, requiring a balance between memory efficiency and preserving node information through node dropping or clustering. In this work, drawing inspiration from bottom-up grammar induction, the authors propose an efficient graph parsing algorithm to infer personalized pooling structures for each graph. The resulting Graph Parsing Network (GPN) leverages discrete assignments from the parsing algorithm, achieving memory efficiency while preserving node information. Experimental results on standard benchmarks demonstrate GPN's superior performance in graph classification tasks compared to state-of-the-art pooling methods, while also showing competitive results in node classification tasks. The paper includes a graph reconstruction task to highlight GPN's ability to preserve node information and assess its memory and time efficiency through relevant tests.

The motivation to adaptively condense different graphs based on their varying structures, specifically edge importance, is sound.
The idea of utilizing a graph parsing algorithm to create an individualized pooling structure for each graph is interesting and reasonable, offering a novel approach to the problem. The experimental results demonstrate that the proposed Graph Parsing Network (GPN) outperforms existing graph pooling methods across both graph-level and node-level tasks, highlighting its effectiveness. Furthermore, the efficiency of GPN in terms of memory usage and computational time, particularly when compared to node-clustering-based graph pooling methods, is noted and appreciated.

I put much more weight on the reviewers giving higher scores, as their reviews were comprehensive and well-argued. In addition, I ignored a negative review without substance.

**Justification For Why Not Higher Score:**

Novelty is limited, similar to some prior work

**Justification For Why Not Lower Score:**

Interesting idea, sound execution, and good experimental results.

---

### Decision · Program_Chairs · 2024-01-16

Accept (poster)